# Two Distinct Genotypes of KPC-2-Producing *Klebsiella pneumoniae* Isolates from South Korea

**DOI:** 10.3390/antibiotics10080911

**Published:** 2021-07-26

**Authors:** Jee Hong Kim, Yun Young Cho, Ji Young Choi, Yu Mi Wi, Kwan Soo Ko

**Affiliations:** 1Department of Microbiology, Sungkyunkwan University School of Medicine, Suwon 16419, Korea; jeehong910@naver.com (J.H.K.); yuncho1008@skku.edu (Y.Y.C.); choiji02@hotmail.com (J.Y.C.); 2Division of Infectious Diseases, Samsung Changwon Hospital, Sungkyunkwan University School of Medicine, Changwon 51353, Korea; yumi.wi@skku.edu

**Keywords:** carbapenemase, KPC, *Klebsiella pneumoniae*

## Abstract

In this study, we investigated the characteristics of KPC-2-producing *Klebsiella pneumoniae* (KP-Kp) isolates from a hospital in South Korea. Among the 37 KP-Kp isolates, two main clones were identified—ST11 and ST307. ST11 isolates showed higher minimum inhibitory concentrations for carbapenems than ST307 isolates. All ST307 isolates were resistant to gentamicin and trimethoprim–sulfamethoxazole, but ST11 isolates were not. However, most tigecycline-resistant or colistin-resistant isolates belonged to ST11. The two KP-Kp clones showed different combinations of *wzi* and K serotypes. Plasmids from ST11 KP-Kp isolates exhibited diverse incompatibility types. Serum resistance and macrophage infection assays indicated that ST11 may be more virulent than ST307. The changes in the main clones of KP-Kp isolates over time as well as the different characteristics of these clones, including virulence, suggest the need for their continuous monitoring.

## 1. Introduction

The emergence of carbapenemase-producing Gram-negative bacteria, including *Klebsiella pneumoniae*, has caused serious therapeutic challenges. In particular, *K. pneumoniae* carbapenemase (KPC) is one of the most threatening carbapenem resistance determinants in clinical settings. Since it was first reported in 2010 in South Korea [1], reports of KPC have been steadily increasing [2,3]. KPC-producing *K. pneumoniae* is difficult to control and often requires a centrally coordinated intervention to combat outbreaks if it is not controlled soon after its emergence [4]. Although more than one hundred different STs have been reported to have the *bla*_KPC_ [5], the increase in KPC producers is associated with the spread of the *K. pneumoniae* ST258 clone in many countries [6,7]. However, other high-risk clones of *K. pneumoniae* have recently been reported to produce KPC, for example, ST11 and ST307 [8,9].

In South Korea, most KPC-producing *K. pneumoniae* isolates belonged to ST11, a single locus variant of ST258 [4]. However, the *K. pneumoniae* clone ST307 has been recently identified as a main KPC producer [2,10]. Although it was reported that *K. pneumoniae* ST307 rarely include virulence factors [11], its virulence was not fully elucidated.

In this study, we identified two main clones of KPC-2-producing *K. pneumoniae* (KP-Kp) isolates from South Korea. We compared the characteristics of the two KP-Kp clones, with respect to antimicrobial resistance, plasmids, serotypes, and virulence.

## 2. Results

During the period of study, we collected 37 KP-Kp isolates. Most KP-Kp isolates were found to belong to ST307 (18 isolates, 48.6%) or ST11 (15 isolates, 40.5%) (Table 1). Two ST789 KP-Kp isolates were also identified. Each isolate showed ST4681 and a single-locus variant of ST11, termed ST11-slv.

Among these, 24 isolates (64.9%) co-produced CTX-M-15 (Table 1). All ST307 KP-Kp isolates produced CTX-M-15, whereas only three ST11 KP-Kp isolates did (*p* < 0.0001). All KP-Kp isolates were resistant to ampicillin, cefepime, ciprofloxacin, aztreonam, piperacillin-tazobactam, and carbapenems (imipenem and meropenem) (Table 1). All ST307 isolates were resistant to gentamicin and trimethoprim–sulfamethoxazole, but only two (*p* < 0.0001) and six (*p* = 0.0001) ST11 isolates were resistant to them, respectively. Four colistin-resistant KP-Kp isolates were identified; three belonged to ST11 and one to ST4681. Among the eight tigecycline-resistant KP-Kp isolates, six belonged to ST307. No ST11 isolates were resistant to tigecycline (*p* = 0.0065).

The distribution of carbapenem MICs was compared between the two main clones—ST11 and ST307 (Figure 1). For both imipenem and meropenem, MICs were higher for ST11 KP-Kp isolates than for ST307 KP-Kp isolates. In particular, KP-Kp isolates with MICs of 64 mg/L for imipenem or >64 mg/L for meropenem were identified only in ST11, except one. Two ST11 isolates exhibited MICs of 64 mg/L or higher for both imipenem and meropenem. They were also resistant to colistin.

Most KP-Kp isolates, except two, possessed plasmids of the FII-incompatibility type (Table 1). However, the subtypes were variable. The plasmids in ST307 isolates showed only three incompatibility types (including IncN6), whereas nine plasmid incompatibility types were identified among the ST11 isolates. All ST307 isolates showed the same *wzi* type and K-serotype (i.e., *wzi*173-KL102), which was also identified in ST4681 isolates (Table 1). ST11 isolates showed two combinations of the *wzi* type and the K-serotype—*wzi*14-K14 and *wzi*50-K51; *wzi*2-K2 and *wzi*18-K18 were identified in the ST11slv and ST789 isolates, respectively. All isolates tested negative in the string test.

The survival rates against human serum were evaluated for each of the two ST11 and ST307 isolates, along with a reference K. pneumoniae strain ATCC 43816 (Figure 2A). The survival rate of ST11 isolate 756 was higher than that of ST11 isolate 297 and two ST307 isolates (*p* = 0.001231 to 0.003317).

We compared the internalization of the bacterial cells by macrophage-like cells between the two main genotypes—ST11 and ST307—of KP-Kp isolates (Figure 2B). The number of intracellular bacteria recovered from the J774A.1 cells reflects the internalization efficiency. The internalization efficiency was similar among the isolates of the same genotype. ST307 KP-Kp isolates showed a significantly higher internalization efficiency than ST11 KP-Kp isolates (*p* = 0.000019).

## 3. Discussion

The first KPC-producing *K. pneumoniae* isolate from South Korea was identified in 2010 [1]. Since then, KPC-2-producing *K. pneumoniae* isolates have been frequently found in South Korea [2,3]. As reported previously [2,8,9,10], our study also showed that KPC was mostly associated with the two *K. pneumoniae* clones ST11 and ST307 in South Korea. We compared the characteristics of these two dominant KP-Kp clones.

ST11 KP-Kp isolates showed higher carbapenem resistance than ST307 KP-Kp isolates. Despite having the same gene (*bla*_KPC-2_), the underlying cause of the difference in carbapenem resistance between the two isolates remains unknown. Nonetheless, it might be caused by the physiological differences in bacterial hosts or the constitution of the plasmids harboring the carbapenemase gene. In addition, loss of porin, such as OmpK35, may be a cause for the difference in carbapenem resistance [12].

In addition to the difference in carbapenem MICs, the two KP-Kp clones showed differences in resistance to some other antibiotics as well. All ST307 KP-Kp isolates were resistant to all antibiotics tested, except for colistin and tigecycline. In contrast, many ST11 KP-Kp isolates were susceptible to gentamicin and trimethoprim–sulfamethoxazole, and the susceptibility was statistically significant. Although *bla*_CTX-M-15_ was identified in all ST307 isolates, only three ST11 isolates possessed it. The plasmid incompatibility types of most ST307 isolates were FIIK7 or FIIK21, but the plasmids of ST11 isolates showed diverse FIIK incompatibility types. However, the plasmids of the *bla*_CTX-M-15_-positive ST11 isolates were of the FIIK7- or FIIK21-incompatibility types. Because two gentamicin-resistant ST11 isolates were positive for *bla*_CTX-M-15_ and possessed FIIK7- or FIIK21-type plasmids, it is plausible that gentamicin resistance genes are carried on the FIIK7- or FIIK21-type plasmids with *bla*_CTX-M-15_. The gentamicin-resistant ST11 KP-Kp isolates might carry the same plasmid as the ST307 KP-Kp isolates. However, direct horizontal transfer of *bla*_KPC-2_-bearing plasmids might not occur frequently between the two clones, because gentamicin-susceptible ST11 KP-Kp isolates possessed plasmid incompatibility types different from those of ST307 isolates.

Notably, although colistin-resistant isolates were identified only in ST11, tigecycline-resistant isolates were found only in ST307. The primary mechanisms underlying colistin and tigecycline resistance, involving modification of lipid A in lipopolysaccharides and overexpression of efflux pump, respectively [13], may not be associated with plasmids. Although plasmid-carried *mcr*-associated with colistin resistance has been identified [14], *mcr* was not identified in the colistin-resistant *K. pneumoniae* isolates in this study. High colistin resistance in a particular clone has been reported in *P. aeruginosa* [15]. However, colistin or tigecycline resistance has not been associated with specific *K. pneumoniae* clones to date. Because colistin and tigecycline are regarded as last-resort antibiotics for the treatment of carbapenem-resistant Gram-negative bacterial infections, it is necessary to investigate whether resistance to them might be better developed in some clones than others.

Virulence assessed by the method of serum resistance and internalization by macrophage-like cells was different between the two main clones of KP-Kp isolates. Although serum resistance was different between the two ST11 isolates, one of the ST11 isolates showed a significantly higher survival rate against human serum than the ST307 isolates. The internalization by macrophage-like cells was lower in the case of the two ST11 isolates than in the case of the two ST307 isolates. Entry of *K. pneumoniae* into macrophage-like cells represents a host defense mechanism and not a bacterial virulence mechanism [16]. Thus, ST11 isolates showing lower internalization rates than ST307 isolates were more virulent. Previously, it was reported that ST307 clones rarely possess virulence factors [11]. Because all KP-Kp isolates included in this study showed negative results in the string test, the high virulence in ST11 may not be due to the hypermucoviscosity or the presence of a capsule. In addition to the capsule, lipopolysaccharide, siderophores, and fimbriae have also been identified as virulence factors in *K. pneumoniae* [17]. Hence, the reason for the difference in virulence between the two clones needs to be explored further.

In this study, we investigated the KP-Kp isolates from South Korea. Although our study has limits in that only a small number of isolates from one hospital were included, we found that there were two main clones that showed different characteristics including antibiotic resistance, plasmids harboring carbapenemase genes, and virulence. Because changes in the prevalent clone of carbapenemase-producing Gram-negative pathogens on a local or nationwide scale may affect the response at the public level, continuous monitoring and analysis of the features of these bacterial strains are required.

## 4. Materials and Methods

### 4.1. Bacterial Isolates

All carbapenemase-producing *K. pneumoniae* isolates included in this study were collected from adult patients at the diverse wards of Samsung Changwon Hospital in South Korea from 2018 to 2019; 13 isolates from sputum, 8 from blood, 7 from rectal swab, 4 from urine, 3 from peritoneal fluid, and 3 from stool. The other six isolates were obtained from bile, genital tract, pus, and wound. The origins of three isolates were unknown. For preliminary species identification, VITEK-2 (bioMérieux) was used, and the results were verified by 16S rRNA gene sequencing. The carbapenemase genes *bla*_KPC_, *bla*_IMP_, *bla*_VIM_, *bla*_NDM_, *bla*_SPM_, and *bla*_SIM_ and an extended spectrum β-lactamase gene *bla*_CTX-M_ were detected by PCR amplification and Sanger sequencing using previously described primers [18,19,20]. The presence of *mcr*, a plasmid-borne colistin resistance gene, was also determined by PCR [21].

### 4.2. Antibiotic Susceptibility Testing

Antibiotic susceptibility was evaluated using the broth microdilution method, following the Clinical and Laboratory Standards Institute guidelines [22]. The minimum inhibitory concentrations (MICs) of 11 antibiotics, including imipenem, meropenem, ampicillin, cefepime, ciprofloxacin, aztreonam, gentamicin, tigecycline, trimethoprim–sulfamethoxazole, piperacillin–tazobactam, and colistin were determined. All antibiotics were purchased from Sigma–Aldrich (St. Louis, MO, USA) except imipenem and meropenem, which were obtained from the manufacturer. CLSI susceptibility breakpoints were employed for all antimicrobial agents except tigecycline. For tigecycline, FDA breakpoints for susceptible (MIC, ≤2 mg/L), intermediate (MIC, 4 mg/L), and resistant (MIC, ≥8 mg/L) were used. All tests were performed in duplicates, and *Escherichia coli* ATCC 25922 and *Pseudomonas aeruginosa* ATCC 27853 were used as controls.

### 4.3. Genotyping and Plasmid Incompatibility Typing

Genotypes of KP-Kp isolates were determined using the multilocus sequence typing (MLST) method, as described previously [23]. Capsular polysaccharide types have previously been identified using *wzi* sequencing [24]. The plasmid incompatibility types of the isolates were identified using the PCR-based replicon-typing method as previously described in the PubMLST website (https://pubmlst.org/organisms/plasmid-mlst/schemes, accessed on 12 April 2021) [25].

### 4.4. String Test

The hypermucoviscosity phenotypes of KP-Kp isolates were determined using the string test [26]. For the test, overcultured bacterial colonies were stretched on blood agar plates using an inoculation loop. Isolates that stretched viscous strings >5 mm were considered hypermucoviscosity-positive isolates [26].

### 4.5. Serum Resistance Assay

For each of the two ST11 and ST307 KP-Kp isolates, the serum resistance assay was performed, as described previously [27,28], with some modifications. Normal human serum (NHS; Innovative Research, Novi, MI, USA) was used to treat mid-log phase bacterial cultures. Heat-inactivated human serum (HIS) was used as control to determine the bactericidal effect of NHS. After 3 h of incubation at 37 °C with shaking, the mixtures were serially diluted and plated on blood agar. The number of colony-forming units (CFUs) that survived after treatment with NHS was compared with the number of CFUs that survived after treatment with HIS. All assays were performed in triplicates, and the results were represented as survival percentages.

### 4.6. Macrophage Infection Assay

The tendency of bacteria to internalize in mammalian cells was determined using the macrophage-like cell line J774A.1, employing a previously described method with slight modifications [29]. The cells were cultured in Dulbecco’s modified Eagle’s medium (DMEM; Welgene, Gyeongsan, Korea) supplemented with 10% (*v*/*v*) fetal bovine serum (FBS; Gibco; Dublin, Ireland) and 1% (*v*/*v*) antibiotic–antimycotic solution (Thermo, Waltham, MA, USA). For the gentamicin protection assay, J774A.1 cells (2.5 × 10^5^ cells) were seeded in a 24-well tissue culture plate (CytoOne 24-well TC plate, USA Scientific, Ocala, FL, USA) the day before bacterial infection. Before infection, the cells were washed three times with Dulbecco’s phosphate buffered saline (DPBS; Welgene, Gyeongsan, Korea) and incubated in DMEM supplemented with FBS for 1 h. Bacterial isolates cultured overnight were inoculated at a ratio of 40 bacteria per macrophage (MOI 40) and incubated for 30 min. Bacterial diluents were enumerated for confirmation. Next, J774A.1 cells were washed three times with DPBS, followed by the addition of DMEM-conditioned medium containing 150 mg/L gentamicin. Subsequently, the cells were incubated for 1 h to eliminate extracellular bacteria. After incubation, the cells were washed three times with DPBS and lysed with 1% Triton X-100. Lysates were sufficiently diluted, and 10 μL of each diluent was dropped onto Luria-Bertani (LB) agar plates for the enumeration of internalized bacterial cells. Internalized bacteria were enumerated by dividing the number of inoculated CFUs by that of cell lysate CFUs. All experiments were performed in triplicates and repeated three times independently.

### 4.7. Statistical Analyses

Statistical analyses were performed using Prism version 8.00 for Windows (GraphPad Software, San Diego, CA, USA). Chi-square tests were used to evaluate the differences in MIC profiles and CTX-M presence between two main clones. The Student’s *t*-test was used to evaluate survival rates among isolates. Statistical significance was set at *p* < 0.05 (*** *p* < 0.0001).

## 5. Conclusions

We identified two main clones of KPC-2-producing *K. pneumoniae* isolates in South Korea, ST11 and ST307. Two main clones showed different characteristics including carbapenem resistance level, *wzi* type and K serotype, plasmid incompatibility, and virulence.

## Figures and Tables

**Figure 1 antibiotics-10-00911-f001:**
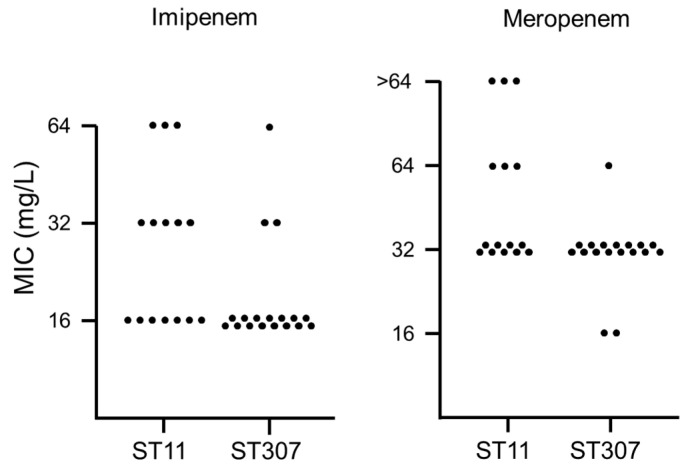
Distribution of the minimum inhibitory concentrations (MICs) of carbapenem antibiotics (imipenem and meropenem) for the two main clones of KP-Kp isolates—ST11 and ST307. The number of dots indicates the number of isolates with the corresponding MIC.

**Figure 2 antibiotics-10-00911-f002:**
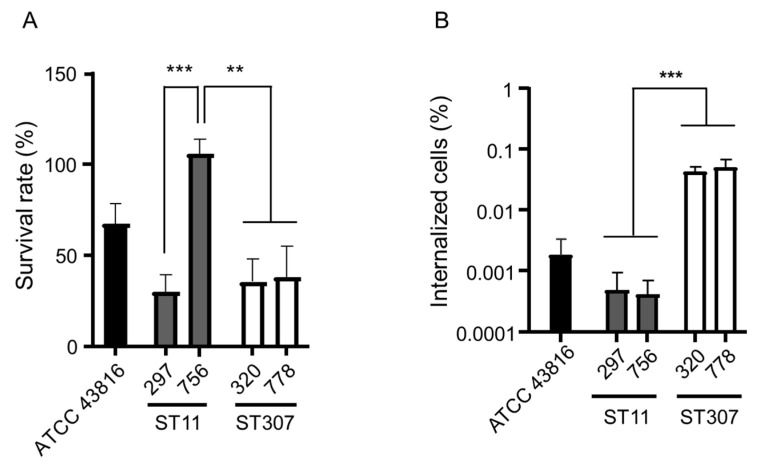
Virulence of the isolates of the two main clones of KP-Kp—ST11 and ST307. (**A**) The result of the serum resistance assay. The survival rate of K. pneumoniae isolates was determined after 3 h of incubation with human serum. Heat-inactivated serum was used as a negative control. (**B**) The result of the macrophage infection assay. The number of internalized bacteria recovered from J774A.1 cells was enumerated. ** *p* < 0.001; *** *p* < 0.0001.

**Table 1 antibiotics-10-00911-t001:** Genotype, antimicrobial resistance, plasmid incompatibility group, *wzi* type, and K-serotype of carbapenemase-producing *K. pneumoniae* isolates investigated in this study.

Genotype	Number of Resistant Isolates (%) ^a^											CTX-M-15	Plasmid	*wzi* Type	K-Serotype
	IMI	MER	COL	AMP	CPM	CIP	GEN	AZT	SXT	P/T	TIG				
ST307 (18)	18 (100)	18 (100)	0	18 (100)	18 (100)	18 (100)	18(100)	18 (100)	18(100)	18 (100)	6 (33.3)	18 (100)	FIIK7 (13) FIIK21 (5) IncN6 (1)	173 (18)	KL102 (18)
ST11 (15)	15 (100)	15 (100)	3 (20.0)	15 (100)	15 (100)	15 (100)	2 (13.3)	15 (100)	6 (40.0)	15 (100)	0	3 (20.0)	FIIK1 (1) FIIK2 (1) FIIK5 (2) FIIK6 (1) FIIK7 (3) FIIK15 (2) FIIK21 (3) FIIK23 (1) FIIY6 (1)	14 (8) 50 (7)	K14 (8) K51 (7)
ST789 (2)	2 (100)	2 (100)	0	2 (100)	2 (100)	2 (100)	0	2 (100)	2 (100)	2 (100)	0	2 (100)	FIIK2 (2)	18 (2)	K18 (2)
ST4681 (1)	1 (100)	1 (100)	1 (100)	1 (100)	1 (100)	1 (100)	1 (100)	1 (100)	1 (100)	1 (100)	1 (100)	1 (100)	FIIK7 (1)	173 (1)	KL102 (1)
ST11slv (1)	1 (100)	1 (100)	0	1 (100)	1 (100)	1 (100)	0	1 (100)	0	1 (100)	0	0	ND ^b^	2 (1)	K2 (1)
Total (37)	37 (100)	37 (100)	4 (10.8)	37 (100)	37 (100)	37 (100)	21 (56.9)	37 (100)	27 (73.0)	37 (100)	8 (21.6)	24 (64.9)			

^a^ IMI, imipenem; MER, meropenem; COL, colistin; AMP, ampicillin; CPM, cefepime; CIP, ciprofloxacin; GEN, gentamicin; AZT, aztreonam; SXT, trimethoprim–sulfamethoxazole; P/T, piperacillin–tazobactam; TIG, tigecycline. ^b^ ND, not determined.

## Data Availability

The data would be available upon request.

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
