# Peer review of "Two Distinct Genotypes of KPC-2-Producing Klebsiella pneumoniae Isolates from South Korea"

_antibiotics, 2021, doi:10.3390/antibiotics10080911_

Round 1
Reviewer 1 Report
The Authors submitted a manuscript regarding the investigation of the characteristics of KPC-2-producing Klebsiella pneumoniae 12 (KP-Kp) isolates from a hospital in South Korea. Although the work is interesting, it needs some changes that I have indicated in the attached file. In particular, the introduction section must be extended (the Authors did not mention all the relevant references) and minor revisions in materials and methods section are required.

Author Response
Reviewer 1
The Authors submitted a manuscript regarding the investigation of the characteristics of KPC-2-producing Klebsiella pneumoniae 12 (KP-Kp) isolates from a hospital in South Korea. Although the work is interesting, it needs some changes that I have indicated in the attached file. In particular, the introduction section must be extended (the Authors did not mention all the relevant references) and minor revisions in materials and methods section are required.
(pdf file)
- The reviewer 1 suggested the extension of Introduction part. In addition, we added some references, as suggested.
“The emergence of carbapenemase-producing gram-negative bacteria, including Klebsiella pneumoniae, has caused serious therapeutic challenges. In particular, K. pneumoniae carbapenemase (KPC) is one of the most threatening carbapenem resistance determinants in clinical settings. Since it was first reported in 2010 in South Korea [1], reports of KPC have been steadily increasing [2,3]. KPC-producing K. pneumoniae is difficult to control and often requires a centrally coordinated intervention to combat outbreaks if is not controlled soon after its emergence [4]. Although more than one hundred different STs have been reported to have the blaKPC [5], the increase in KPC producers is associated with the spread of K. pneumoniae ST258 clone in many countries [6,7]. However, other high-risk clones of K. pneumoniae have recently been reported to produce KPC, for example, ST11 and ST307 [8,9].
In South Korea, most KPC-producing K. pneumoniae isolates belonged to ST11, a single locus variant of ST258 [4]. However, the K. pneumoniae clone ST307 has been recently identified as a main KPC producer [2,10]. Although it was reported that K. pneumoniae ST307 rarely include virulence factors [11], its virulence was not fully elucidated.” (Line 26-40 in the revised manuscript)
- Line 25. ‘gram’ – We did not revise it because small capital (gram) should be used when referring the staining method. Large capital (Gram) refers to the name of the person who developed the method.
- Through the text, we italicized all bacterial names such as K. pneumoniae, Pseudomonas aeruginosa, and E. coli.
- In Table 1. “CTX” in “CTX-M-15”. In general, we know that CTX in CTX-M-15 does not express what it stands for.
- Line 151-152. The Authors must indicate Ethical approval ID/number. From which samples bacterial strains were isolated?
- This study used only bacterial isolates, but didn’t use the information patients. Thus, Ethical approval was unnecessary.
- We mentioned the origin of bacterial strains.
“13 isolates from sputum, 8 from blood, 7 from rectal swab, 4 from urine, 3 from peritoneal fluid, and 3 from stool, respectively. The other six isolates were obtained from bile, genital tract, pus, and wound. The origin of three isolates was unknown.” (Line 173-175 in the revised manuscript)
- Line 161-163. Where were the antibiotics purchased?
- We indicated where we purchased the antibiotics.
“All antibiotics were purchased from Sigma-Aldrich (St. Louis, MO, USA) except imipenem and meropenem, which were obtained from the manufacturer.” (Line 187-189 in the revised manuscript)
- Line 194-196. We included units.
- Line 196. “24-well tissue culture plate”
- We indicated microplate type and its producer.
“24-well tissue culture plate (CytoOne 24-well TC plate, USA Scientific, Ocala, FL, USA)” (Line 223 in the revised manuscript)
Reviewer 2 Report
In this study, the authors investigated the characteristics of KPC-2-producing Klebsiella pneumoniae (KP-Kp) isolates from South Korea and compared the characteristics of the two KP-Kp clones with respect to antimicrobial resistance, plasmids, serotypes, and virulence. This is an interesting work and my only concern is the statistical analyses. The authors claimed that Student's t-test, one-way ANOVA with Tukey’s multiple comparisons test, nonparametric Kruskal–Wallis test accompanied by Dunnett's multiple comparison test, and Chi-square test were used to evaluate the differences. This statement is meaningless. Why is it necessary to use so many different statistical analyses method in this work? Please include the detailed statistical analyses method in the relevant place in the result section and the caption of Figure 2.
Author Response
In this study, the authors investigated the characteristics of KPC-2-producing Klebsiella pneumoniae (KP-Kp) isolates from South Korea and compared the characteristics of the two KP-Kp clones with respect to antimicrobial resistance, plasmids, serotypes, and virulence. This is an interesting work and my only concern is the statistical analyses.
- The authors claimed that Student's t-test, one-way ANOVA with Tukey’s multiple comparisons test, nonparametric Kruskal–Wallis test accompanied by Dunnett's multiple comparison test, and Chi-square test were used to evaluate the differences. This statement is meaningless. Why is it necessary to use so many different statistical analyses method in this work? Please include the detailed statistical analyses method in the relevant place in the result section and the caption of Figure 2.
- We revised the part of “Statistical analyses”
“Statistical analyses were performed using Prism version 8.00 for Windows (GraphPad Software, San Diego, CA, USA)., Chi-square test were used to evaluate the differences in MIC profiles and CTX-M presence between two main clones. Student's t-test was used to evaluate survival rates among isolates. Statistical significance was set at P < 0.05 (***, P < 0.0001).“ (Line 238-242 in the revised manuscript)
Reviewer 3 Report
The article submitted by Kim J et al describes the main differences between two clones of KPC-2- producing Klebsiella pneumoniae from South Korea, ST-11 and ST-307. This study contains some interesting results; however, there are a few points to comment on.
Line 38: The number of isolates included in the study (n=37) should be increased, adding isolates from previous years or recovered in other hospitals, in order to obtain more statistical significance.
Lines 40 and 41: “Two ST789 KP-Kp isolates were also identified; each isolate showed ST4681 and a single locus variant of ST307, termed ST-11-slv”. This sentence is not clear. As shown in Table 1, there are four isolates that do not belong to either ST307or ST11: two ST789, one ST4681, and one ST11-slv. Please, describe this point more clearly in the text.
Lines 45-49: It would be interesting to test at least the main determinants of resistance (amynoglicoside modifying enzimes, mcr) by PCR. Although authors indicate (line 122) that mcr was not identified in the colistin-resistant isolates, no information on how it was performed is described in “Materials and methods”
Line 52: The loss of porins (mainly OmpK35) could contribute to higher carbapenems MICs. Authors should discuss this point.
Line 61: ST307slv? It does not appear in Table 1.
Table 1: Authors may consider including MICs range.
Bacterial isolates: It is important that the bacterial isolates included in this study were not clonally related (belonging to the same patient or the same outbreak). Please, explain the inclusion criteria.
Author Response
Reviewer 3
The article submitted by Kim J et al describes the main differences between two clones of KPC-2- producing Klebsiella pneumoniae from South Korea, ST-11 and ST-307. This study contains some interesting results; however, there are a few points to comment on.
- Line 38: The number of isolates included in the study (n=37) should be increased, adding isolates from previous years or recovered in other hospitals, in order to obtain more statistical significance.
- Thank you for important advice of reviewer 3. We also know that the number of isolates included in our study was small. However, our results are meaningful in spite of such small number of isolates. We mentioned the limit of our study in the revised manuscript.
“Although our study has limits in that only a small number of isolates from one hospital was included,” (Line 161-162 in the revised manuscript)
- Lines 40 and 41: “Two ST789 KP-Kp isolates were also identified; each isolate showed ST4681 and a single locus variant of ST307, termed ST-11-slv”. This sentence is not clear.
- We revised the sentences. There was a typo error (ST11 not ST307).
“Two ST789 KP-Kp isolates were also identified. Each one isolate showed ST4681 and a single-locus variant of ST11, termed ST11-slv.” (Line 46-48 in the revised manuscript)
- As shown in Table 1, there are four isolates that do not belong to either ST307 or ST11: two ST789, one ST4681, and one ST11-slv. Please, describe this point more clearly in the text.
- We revised the text, as replied in #2.
“Two ST789 KP-Kp isolates were also identified. Each one isolate showed ST4681 and a single-locus variant of ST11, termed ST11-slv.” (Line 47-49 in the revised manuscript)
- Lines 45-49: It would be interesting to test at least the main determinants of resistance (amynoglicoside modifying enzimes, mcr) by PCR. Although authors indicate (line 122) that mcrwas not identified in the colistin-resistant isolates, no information on how it was performed is described in “Materials and methods”
- mcr is a mobile colistin resistance gene in plasmid. The detection method of mcr was described in “Materials and methods”.
“The presence of mcr, a plasmid-borne colistin resistance gene, was also determined by PCR [21]” (Line 180-181 in the revised manuscript)
- Line 52: The loss of porins (mainly OmpK35) could contribute to higher carbapenems MICs. Authors should discuss this point.
- As suggested, we discussed the effect of loss of porin.
“In addition, loss of porin such as OmpK35 may be a cause of difference in carbapenem resistance [12].” (Line 117-118 in the revised manuscript)
- Line 61: ST307slv? It does not appear in Table 1.
- It’s our typo. ST11slv is right, we revised it.
- Table 1: Authors may consider including MICs range.
- Inclusion of MIC range (in each ST) would make the table too complicated. We think that there is no trouble to understand it even if the MIC range was not presented.
- Bacterial isolates: It is important that the bacterial isolates included in this study were not clonally related (belonging to the same patient or the same outbreak). Please, explain the inclusion criteria.
- We included all carbapenemase-producing K. pneumoniae isolates collected from the hospital during the period.
“All carbapenemase-producing K. pneumoniae isolates included in this study were collected from patients at the Samsung Changwon Hospital in South Korea from 2018 to 2019.” (Line 171-173 in the revised manuscript)
Round 2
Reviewer 3 Report
Accepted
Author Response
1. The enrolled patients were only adults?
2. In which wards They were admitted?
- Yes. all patients were adults. We included this information.
- Regrettably, we had no information on that. As a result of the inquiry, I received a reply that it was a strain from a wide variety of wards. But just because he(she) can't provide that information, we can't include it specifically.
"All carbapenemase-producing K. pneumoniae isolates included in this study were col-lected from adult patients at the diverse wards of Samsung Changwon Hospital in South Korea from 2018 to 2019" (In Materials and methods)
